# Plants in Menstrual Diseases: A Systematic Study from Italian Folk Medicine on Current Approaches

**DOI:** 10.3390/plants13050589

**Published:** 2024-02-22

**Authors:** Rosalucia Mazzei, Claudia Genovese, Angela Magariello, Alessandra Patitucci, Giampiero Russo, Giuseppe Tagarelli

**Affiliations:** 1Institute for Agricultural and Forest Systems in the Mediterranean, National Research Council, Via Cavour 4-6, 87036 Rende, Italy; 2Institute for Agricultural and Forest Systems in the Mediterranean, National Research Council, Via Empedocle, 58, 95128 Catania, Italy; 3Family Counseling Center (ASP Cosenza), 87036 Rende, Italy

**Keywords:** menstrual diseases, PMS, amenorrhea, dysmenorrhea, menstrual disorders, folk medicine, plant remedies, ethnomedicine, plant-based drugs

## Abstract

**Background:** Plant-based remedies have been used since antiquity to treat menstrual-related diseases (MD). From the late nineteenth to the early to mid-twentieth century, Italian folk remedies to treat “women’s diseases” were documented in a vast *corpus* of literature sources. **Aim:** The purpose of this paper is to bring to light the plant-based treatments utilized by Italian folk medicine to heal clinical manifestations of premenstrual syndrome (PMS), dysmenorrhea, amenorrhea and menstrual disorders in an attempt to discuss these remedies from a modern pharmacological point of view. Moreover, we compare the medical applications described by Hippocrates with those utilized by Italian folk medicine to check if they result from a sort of continuity of use by over two thousand years. **Results:** Out of the 54 plants employed in Italian folk medicine, 25 (46.3%) were already documented in the *pharmacopoeia* of the *Corpus Hippocraticum* for treating MD. Subsequently, a detailed search of scientific data banks such as Medline and Scopus was undertaken to uncover recent results concerning bioactivities of the plant extracts to treat MD. About 26% of the plants used by Italian folk medicine, nowadays, have undergone human trials to assess their actual efficacy. At the same time, about 41% of these herbal remedies come back to in different countries. **Conclusions:** Active principles extracted from plants used by Italian folk healers could be a promising source of knowledge and represent strength candidates for future drug discovery for the management of MD.

## 1. Introduction

Menstruation, also called “menses” or “period”, represents the monthly flow of blood during the years between puberty and menopause. There are some women that go through their monthly periods easily with their periods being punctual and no inconvenience. On other hand, menstrual diseases represent a prominent cause for gynecological consultations globally, and diverse definitions and classifications are used to characterize this condition. Menstrual experiences include a series of physical and/or emotional symptoms just before and during menstruation that may disrupt the quality of life: heavy bleeding, missed menstruations and unmanageable mood swings.

Menstrual diseases (MD) include the following: dysmenorrhea, when there are painful cramps during menstruation; premenstrual syndrome (PMS) with physical and psychological symptoms occurring prior to menstruation; menorrhagia with heavy bleeding, including prolonged menstrual periods or excessive bleeding during a normal-length period; metrorrhagia with bleeding at irregular intervals, particularly between expected menstrual periods; amenorrhea with absence of menstruation; and oligomenorrhea with infrequent menstrual periods.

In accordance with a systematic review conducted by the World Health Organization, the incidence of dysmenorrhea among women of reproductive age ranged from 16.8% to 81.0%, with severe symptoms reported in 12–14% of cases [1].

The prevalence of PMS has been reported in 30–40% of the reproductive female population [2].

Moreover, a prevalence ranging from 14% to 25% has been observed among women experiencing irregular menstrual cycles. This variability encompasses cycles that are either shorter or longer than the typical duration, exhibit abnormal flow in terms of heaviness or lightness, or are accompanied by other issues such as abdominal cramps [3].

Therefore, it is unsurprising that menstruation has garnered considerable attention throughout human history. Speculation concerning the underlying biological processes has led to numerous hypotheses regarding the purpose and characteristics of menstrual blood. Various cultures have developed distinct beliefs and practices, each associated with its own set of rituals in relation to menstruation.

Herbal remedies have been used to handle MD for centuries in many cultures. In the Mediterranean area the “Corpus Hippocraticum” (a collection of 62 medical works written between the 5th century BCE and the 2nd century AD) that were ascribed to Hippocrates (460–377 BC), represents the early medical rational approach that looks to the disease as a natural phenomenon linked to the body and not to supernatural forces [4]. In the opinion of Hippocratic physicians, menstruation was regarded as crucial to the general health of women, and every anomaly was addressed through the administration of medicinal plants aimed at either promoting, inhibiting, or regulating the menstrual process [5].

In our preceding studies, we explored plant-based remedies used by traditional Italian medicine for addressing malaria [6], epilepsy [7], headaches [8], and bacterial skin disorders [9]. These papers paved the way for the investigation of the plant-based remedies used by Italian folk medicine for the management of MD during the latter part of the nineteenth century to the early to mid-twentieth century. For this reason, we analyzed published pharmacological data related to the identified herbal remedies, evaluating their potential use to help women in menstrual cycle management. Therefore, a review of scientific literature sources was carried out for retrieving therapeutic practices recommended in popular Italian medicine to heal MD such as dysmenorrhea, amenorrhea, PMS, and menstrual disorders (irregular cycles, menorrhagia, metrorrhagia, and oligomenorrhea).

The objective of this review is to summarize information regarding MD herbal remedies recommended during the latter half of the 19th century up to the onset of the “economic boom” in the 1950s. The starting point of this timeframe aligns with the period in which the study of folk traditions, as remedies to heal several diseases, arose and spread out with an appropriate methodology [10]. The conclusion of the 1950s corresponds to the period of economic prosperity, marked by a significant rural-to-urban migration, resulting in a notable shift where, for the first time in Italy, a greater number of individuals were employed in industrial sectors than in agriculture [11] transforming Italy from a mainly rural to a modern country [12].

## 2. Materials and Methods

This review was designed and performed in order to gather information regarding plant-based remedies for MD in traditional Italian folk medicine juxtaposed with contemporary scientific knowledge. Moreover, herbal remedies of Italian folk medicine were compared with medical applications described in the *Corpus Hippocraticum* to check if they result from a sort of cultural heritage of use by over two thousand years.

The literature sources regarding Italian folk medicine were searched exploring the National Library Service website of the Italian Libraries Network. For this aim, “medicina popolare” (folk medicine), “rimedi popolari” (popular remedies), “usi e costume” (usages and customs), “tradizioni popolari” (traditions), “cultura tradizionale” (traditional knowledge), and “mestruazioni”, “malattie delle donne”, “segrete cose delle donne” (mestruation) were used as key words. In this way, we collected about one hundred sources (books and journal articles), written by anthropologists, physicians, ethnographers, folklorists, and scholars of local history.

The plants have been identified to the species level whenever the text mentioned the associated Latin binomial or whenever the vernacular Italian name was unambiguous (e.g., aglio = *Allium sativum* L). In cases where the vernacular name was not sufficient for a specific identification, the genus was attributed (e.g., oak, *Quercus* spp.). The plant species are reported according to the WFO Plant List website www.worldfloraonline.org accessed on 8 February 2024) [13] while the families follow the Angiosperm Phylogeny Group 4 [14].

Subsequently, the identified plant taxa were used for a systematic literature research, performed from 2019 to 2023 on Medline and Scopus databases, in order to retrieve the use of herbal drugs for the treatment of MD. All scientific studies were searched, matching the name of medical plants used by Italian folk medicine with the following keywords: amenorrhea, oligomenorrhea, dysmenorrhea, menstruations, metrorrhagia, menorrhagia, premenstrual syndrome, and menstrual disorders. At last, we grouped the collected medicinal plants into four categories, following the cohorts established by Harlow and Campbell [15] for menstrual morbidity, and the definition of PMS provided by Gnanasambanthan and Datta [16]: (1) dysmenorrhea; (2) PMS; (3) amenorrhea; and (4) menstrual disorders, which encompass irregular cycles, such as prolonged menstrual flow and excessive, heavy bleeding (menorrhagia/polymenorrhagia), or delayed, infrequent menses (oligomenorrhea).

## 3. Results

### 3.1. Corpus Hippocraticum’s Data

Out of the 54 plants employed in Italian folk medicine during the late nineteenth to the early to mid-twentieth century, 25 (46.3%) were already documented in the *pharmacopoeia* of the *Corpus Hippocraticum* for treating MD.

### 3.2. Rituals and Magical Practices

In various geographical areas and across diverse historical epochs, a prevailing notion exists whereby menstruating women are perceived as possessing inherent impurity and concurrently as agents of pollution, with their menstrual blood regarded as potential hazards. Due to these beliefs, Italian folk healers used some plants as amulets, for example, putting Southern maidenhair fern aerial parts or oleander leaves in the shoes, to treat amenorrhea [17]. In other cases, cypress leaves were held on the neck to regulate menstruation or applied on the skin with southern wormwood leaves to treat amenorrhea [18]. Saffron dried stigma enveloped in a piece of paper were put on the stomach [19] and white water lily leaves applied on the pubis [18] were used to regulate menstruation and to resolve metrorrhagia, respectively. Finally, the use of plants as herbal drugs was often accompanied with the recitation of secret words [20]. In some cases, the effect of lavender in regulating menstruation was enhanced if harvested during the Night of Saint John [18].

### 3.3. Botanical Analysis and Therapeutic Applications

The collected data pointed out 54 plant species belonging to 28 families. Out of these, the most cited were Asteraceae (Compositae) (13.0%), Lamiaceae (13.0%), Apiaceae (7.4%), and Poaceae (7.4%). The study has identified 82 plant-based remedies. These treatments encompass twelve recipes, each of which incorporates, in addition to an herbal remedy, at least one of the following elements: plants, animal parts, or minerals. Remedies were administered by oral routes (67.9%), applied to the skin (20.2%), and not specified (11.9%). The most common way of administration was decoctions (48.8%). The parts of plants used are the following: leaves (36.4%), flowers (16.7%), aerial parts (14.6%), fruits (7.3%), roots (6.2%), seeds (5.3%), bark (1.1%), resin (1.1%), whole plant (1.1), and not specified (10.4%) (Table 1).

We found an intriguing proportion (66.7%) of plants utilized in Italian folk medicine, from the latter half of the nineteenth to the early mid-twentieth century to treat MD, in the present scientific literature. In fact, 25.9% of plants exhibit effective properties in human trials for the management of MD. (Table 2). On the other hand, 40.8% of plants have been widely used by women globally, as non-conventional medicine (Table 3). Finally, 33.3% of the plants are not observed in the present scientific literature for the management of MD.

## 4. Discussion

Natural products play a crucial role in the discovery of new bioactive compounds for the development of drugs to treat human diseases. In fact, over half of all approved small-molecule drugs have a structure originally derived from natural products. Moreover, a substantial number of natural products are currently under investigation in clinical trials [116]. Likewise, treatments sourced from plants and animals have served as an inexhaustible source of remedies since ancient times, with their knowledge passed down over centuries through traditional medicine [117]. According to De Vos [118], ancient natural remedies originating from the Mediterranean region had a remarkable presence in the European *materia medica*, persisting well into the 19th century.

This study highlights that 46.3% of the plants used in Italian folk medicine from the late nineteenth to the early to mid-twentieth century to treat MD, appear to represent the continuation of a medical tradition spanning about two thousand years.

Recent research suggests that the knowledge of folk medicine has its roots in the *pharmacopeia* of ancient Greece and Rome. [6,7,8,119,120]. However, the challenge lies in definitively identifying the specific plants (and diseases) being referred to. Despite these difficulties, an interdisciplinary approach that incorporates philological, botanical, zoological, anthropological, and pharmacological perspectives may offer the best chance of identifying natural remedies and diseases described in ancient texts.

According to Aliotta and colleagues [121], many scholars who have studied Hippocratic texts generally agree on identifying the plant-based prescriptions mentioned in the *Corpus Hippocraticum*. This consensus is corroborated by Riddle [122], who notes that only 11 out of 257 plants in the *Corpus Hippocraticum* cannot be identified with certainty.

From a pharmacological perspective, the plants utilized in Italian folk medicine for treating menstrual diseases contain essential components such as vitamins, microelements, and metabolites like flavonoids and terpenoids. These constituents exhibit a wide range of anti-inflammatory, anti-nociceptive, and analgesic activities, achieved through inhibitory effects against crucial pathways, including nuclear factor kappa-light-chain-enhancer (NF-ĸB), nitric oxide (NO), cyclooxygenase-2 (COX-2), lipoxygenases (LOX), tumor necrosis factor-α (TNF-α), prostaglandins (PGs), and serotonergic dysregulation. These pathways could play a pivotal role in triggering various types of menstrual diseases as shown in recent human trials concerning dysmenorrhea, PMS, amenorrhea, and menstrual disorders.

Several human trials demonstrated the efficacy of plant extracts in relieving MD symptoms.

For example, *Foeniculum vulgare* Mill. was used in clinical human trials to treat dysmenorrhea and PMS. Fennel extracts showed more effective results than the placebo in pain relief due to the ability to inhibit contractions induced by oxytocin and PGE2 as shown in uterus rats [40].

*Pimpinella anisum* L. is capable of decreasing the intensity of PMS symptoms and improving women’s health. Anise works as a selective moderator of estrogenic receptors, which show agonist/antagonist effects [47].

The oil extracted from the seeds of the *Borago officinalis* L. plant is a rich source of ɣ-linoleic acid (GLA). The underlying mechanism of action of GLA is believed to result from its downregulation of PGE2 production, which takes place by a rapid conversion of GLA to dihomo-ɣ-linolenic acid (DGLA). This conversion increases PGE1 production, and consequently raises intracellular cyclic adenosine monophosphate (cAMP) levels, which in turns inhibits phospholipase, thus limiting the release of arachidonic acid (AA). These metabolites could be effective in the treatment of the physical and emotional symptoms of PMS [52].

*Boswellia* spp. appears to have the potential as a beneficial complementary therapy for addressing heavy menstrual bleeding. The anti-inflammatory properties of boswellic acid and its derivatives are thought to arise from their ability to inhibit enzymes such as LOX, nitric oxide synthase, COX-2, and PGs, as indicated by Eshaghian and colleagues [53].

The extracts of *Valeriana officinalis* L. help with the management of dysmenorrhea, inhibiting contractions triggered by cellular depolarization by opening potassium channels and blocking calcium channels. The opening of potassium channels results in a decrease in intracellular calcium concentration, leading to muscle relaxation, as highlighted by Mirabi and colleagues [54]. Additionally, valerian root extract helps to prevent smooth muscle contraction in the uterus during menstruation by inhibiting the release of PGs, ultimately contributing to pain relief in women, as suggested by Moghadam and colleagues [55].

In other human trials, there was a significant difference noted between the use of *Matricaria chamomilla* L. and mefenamic acid, with chamomile demonstrating a greater reduction in primary dysmenorrhea pain. While the exact cause of PMS remains unknown, it is plausible that many of the mediators contributing to PMS symptoms are PGs. This is supported by the anti-prostaglandin properties of chamomile. Additionally, chamomile exerts anti-inflammatory and sedative effects through constituents such as chamazulene and alpha-bisabolol, as highlighted by Dadfar [49].

Effective treatments for relieving PMS include serotonin re-uptake inhibitors, suggesting an underlying serotonergic dysregulation. It is worth noting that *Hypericum perforatum* L. significantly increases serotonin levels in the brain, leading to an upregulation of serotonin 5-HT1A and 5-HT2A receptors, as demonstrated by Stevinson and Ernst [56]. Additionally, there is a potential connection between heightened production of proinflammatory cytokines and the onset of PMS symptoms. During the luteal phase, cytokines associated with hallmark PMS symptoms have been observed to escalate. Notably, St. John wort extracts have demonstrated the ability to inhibit the production of proinflammatory cytokines [59].

The action of metabolites of *Crocus sativus* L. could improve the management of MD. The anxiolytic effects of saffron were distinctly revealed through alterations in cortisol and estrogen, modulating steroid hormone levels via an olfactory sensory perception mechanism. Furthermore, saffron has provided insights into the impact of sex hormones on serotonin uptake, binding, turnover, and transport [60]. Moreover, certain constituents like flavonoids and carotenoids in saffron exhibit antioxidant properties. By capturing free radicals such as oxygen and superoxide radicals, they inhibit the production of PGs, as highlighted by Beiranvand and colleagues [62].

*Lavandula angustifolia* Mill. Could act postsynaptically modulating the activity of cAMP. A decrease in cAMP levels is correlated with sedative effects. Additionally, linalool, which represents one of the components of lavender oil, has been identified as an inhibitor of glutamate binding, suggesting potential sedative effects [67]. Moreover, according to Raisi Dehkordi and colleagues [68], linalyl acetate has narcotic actions and linalool acts as a sedative probably related to antimuscarinic activity and/or (Na or Ca) channel blockade. Finally, linalool is effective in suppressing the release of PGs, which are responsible for inducing contractions in the uterine muscles. The heightened secretion of PGs, in fact, may lead to intensified myometrial contractions, uterine ischemia, and subsequent cramping accompanied by pelvic pain [72].

The alcoholic extract of *Salvia Rosmarinus* Spenn. leaves, containing carnosic acid and carnosol, has the capability to diminish prostaglandin production by lowering interleukin β (IL β), TNF-α, and COX-2 levels. Rosemary capsules exhibit a comparable efficacy to mefenamic acid capsules in reducing menstrual bleeding and alleviating primary dysmenorrhea [73].

*Salvia officinalis* L. exhibits anti-inflammatory, anti-anxiety, and antioxidant properties attributed to its constituents, including rosmarinic acid, carnosic acid, gallic acid, flavonoids, diterpenes, and phenolic acids. The phytoestrogenic compounds present in sage may act as alleviating agents through neurochemical receptors in the central nervous system, influencing learning, anxiety, and the hypothalamus–pituitary axis. Supplementation with phytoestrogenic compounds could potentially have significant effects on both brain function and behavior. The anti-inflammatory and antioxidant attributes of sage, due to rosmarinic acid, carnosic acid, and caffeic acid, along with the anti-prostaglandin effects of carnosol and carnosic acid, may contribute to the reduction of physical and psychological symptoms associated with PMS [74].

The impact of *Vitex agnus-castus* L. on PMS has been likened to that of the corpus luteum. The mechanism of action of vitex may be linked to the modulation of stress-induced prolactin secretion through dopamine, without a direct influence on luteinizing hormone or follicle-stimulating hormone. Additionally, its role may involve binding to opioid receptors, endorphins, and neuroactive flavonoids, as suggested by Schellenberg and colleagues [75].

The beneficial impact of vitamin B6 in alleviating symptoms such as depression, anxiety, irritability, and breast tenderness, as well as its superior effectiveness compared to placebos, underscores the role of B vitamins in wheat germ (*Triticum* spp.) in regulating psychological well-being and addressing mood imbalances. Particularly, in PMS-related depression, the justification for this lies in the facilitation of serotonin production, in the modulation of tryptophan metabolism, and in the action of pyridoxine-phosphate (the active form of vitamin B6) which is involved in the synthesis of numerous neurotransmitters. Furthermore, the zinc content in wheat germs may interact with neurotransmitters, influencing the psychological balance and symptom reduction. Additionally, according to Ataollahi and colleagues, the presence of iron in wheat germ is crucial as a cofactor for tryptophan hydroxylase, with reported involvement in the metabolism of serotonin and ɣ-aminobutyric acid (GABA) [89].

Finally, the relief of dysmenorrhea through the use of *Salix* spp. is likely attributed to its inhibition of the COX and LOX pathways in prostaglandin synthesis, as suggested by Raisi Dehkordi and colleagues [90].

The investigation of the recent literature brought to light many ethnographic surveys about plant-based remedies used in different countries.

Out of fifty-four plants used by Italian folk medicine, from the late nineteenth to the early to mid-twentieth century, to treat “women’s diseases”, twenty-two (40.8%) are employed in recent surveys that documented the traditional use of plants to manage MD. Herbal medicine has been extensively utilized by women worldwide, with a growing recognition of the essential role it plays in both disease treatment and health maintenance. Even in the modern era, certain health issues persist among women that may not be effectively addressed by conventional medicine. In regions lacking access to modern medical treatments, the use of herbal medicine remains vital as an alternative or supplement for women’s health, contributing to the enhancement of their overall well-being [95].

The exploration of traditional knowledge, nearly forgotten during the 20th century, has garnered increasing interest in the rational design and development of drugs. This interest is not adequately reflected in regulatory procedures or the concept of evidence-based medicine (EbM). The EbM concept categorizes knowledge sources into five or six categories, ranking meta-analyses of prospective, double-blind human trials as the highest form. Preparatory to these studies are those conducted in vivo and in vitro. However, this scale of values largely overlooks the value of traditional use and the highly esteemed knowledge basis derived from historical research. Moreover, the recipes employed in Italian folk medicine present a distinct characteristic. Each of these recipes include not only the use of several plants, for example as the infusion of seven roots of Southern maidenhair fern, chamomile, mallow, lettuce, chicory, celery and fennel to regulate menstruation [19], but also the use of animal parts, or minerals. In fact, pigeon dried blood (3 drams) was employed with a half dram of powder of chamomile flowers (1 once), savin juniper flowers (1 once), saffron flowers (2 dram), incense seeds (1 crap), blessed in a half dram of wine, to treat amenorrhea [30]. Finally, antimony was mixed in a decoction of savin juniper and Southern maidenhair fern leaves to regulate menstruation [19]. This feature represents a specific limitation of the current literature-based approach. The existing scientific methodology often neglects the simultaneous analysis of polyherbal preparations. In fact, there is a lack of potential synergistic effects that may arise from the combination of multiple plants. Furthermore, the prevalence of traditions in geographically separate regions is another important criterion. The more clusters exist that follow the same traditional treatment method independently of each other, the more valid the tradition and its evidence can be seen to be [123]. Some of the plants used in the past by Italian folk medicine appear in recent ethnographic surveys describing their use to treat amenorrhea, dysmenorrhea, and menstrual disorders, in many areas of the world. For example, *Petroselinum crispum* (Mill.) Fuss is used in Italy, Brazil, Dominican Republic, and Morocco [91,93,94]; *Hedera elix* L. is employed in Romania and Italy [95,96]; *Artemisia absinthium* L. is utilized in Iran, Brazil, Dominican Republic [93,94,97]; *Malva sylvestris* L. is used in Brazil, Pakistan, and Italy; *Ruta graveolens* L. is employed in Brazil, Dominican Republic, Italy, Trinitad and Tobago [93,94,95,107,112]; and *Urtica dioica* L. is used in Italy, Pakistan, and Serbia [91,99,114] (Table 3). All of this leads us to draw attention to the above highly promising species and to candidate them to deeply experimental studies. They, in fact, have been commonly used for medicinal purposes, starting from Hippocratic treatises to Italian folk medicine and, finally, in recent ethnographic surveys spanning centuries and various countries.

## 5. Conclusions

In conclusion, this paper identifies plants of interest for future research and advocates for a more critical and innovative approach to studying traditional knowledge. The rich history of “therapeutic wisdom” embedded in traditional use provides a valuable foundation for uncovering potential drugs of the future, urging researchers to explore synergistic mechanisms and consider the intricate interplay of multiple plants in their investigations.

## Figures and Tables

**Table 1 plants-13-00589-t001:** Herbal drugs recommended in folk Italian medicine to treat menstrual diseases between the latter half of the nineteenth century and the early to mid-twentieth century and mentioned to serve the same purpose by *Corpus Hippocraticum*.

Family/Scientific/Common Name	Italian Name	Mode of Administration/Part Plant Used/Menstrual Diseases	References	Corpus Hippocraticum	References
**Adoxaceae**					
*Sambucus nigra* L. (Elderberry)	Ebolo	Use elderberry root to treat amenorrhea	[20]	Diseases of Women I, 47, 66; Diseases of Women II, 193–194	[21]
**Apiaceae**					
*Apium graveolens* L. (Celery)	Sedano	Drink a decoction of Southern maidenhair fern, chamomile, mallow, lettuce, chicory, celery and fennel roots to regulate menstruation	[19]	Diseases of Women II, 113, 193, 195, 282	[21]
		Drink a decoction of celery leaves to treat menorrhagia	[18]		
*Foeniculum vulgare* Mill. (Fennel)	Finocchio	Drink fasting a decoction of parsley root, fennel, mallow leaves and chamomile flowers, early in the morning, to regulate menstruation	[19]	Diseases of Women II, 113, 215	[21]
		Drink a decoction of Southern maidenhair fern, chamomile, mallow, lettuce, chicory, celery and fennel roots to regulate menstruation			
*Petroselinum crispum* (Mill.) Fuss (Parsley)	Prezzemolo	Drink an infusion of parsley plant (including root) to treat amenorrhea	[22]	Diseases of Women II, 113, 193, 195, 282	[21]
		Drink fasting a decoction of parsley root, fennel, mallow leaves and chamomile flowers, early in the morning, to regulate menstruation	[19]		
		Drink a decoction of parsley leaves to treat metrorrhagia	[18]		
*Pimpinella anisum* L. (Anise)	Anice	Drink a decoction of anise seeds to treat amenorrhea	[18]	Diseases of Women I, 74; Diseases of Women II, 195, 196; Nature of Women, 32	[21,22,23]
		Eat a salad of fresh vegetables and anise seeds to regulate menstruation	[24]		
**Apocynaceae**					
*Nerium oleander* L. (Oleander)	Oleandro	Hold under shoes oleander leaves to treat amenorrhea	[17]		
Araliaceae					
*Hedera helix* L. (Common Ivy)	Edera	Drink a decoction of common ivy leaves to regulate menstruation	[24]	Diseases of Women II, 193–194	[21]
**Asteraceae**					
*Artemisia abrotanum* L. (Southern Wormwood)	Abrotano	Hold on the skin southern wormwood and cypress leaves to treat amenorrhea	[18]	Nature of Women, 109	[23]
		Use southern wormwood leaves, generically, to treat menstrual disorders			
*Artemisia absinthium* L. *Artemisia pontica* L. (Wormwood)	Assenzio	Use wormwood leaves, generically, to treat amenorrhea	[18]	Nature of Women, 32, 109	[23]
*Cichorium intybus* L. (Blue daisy)	Cicoria	Drink an infusion of Southern maidenhair fern, chamomile, mallow, lettuce, chicory, celery and fennel roots to regulate menstruation	[19]		
*Lactuca sativa* L. (Lettuce)	Lattuga	Drink an infusion of Southern maidenhair fern, chamomile, mallow, lettuce, chicory, celery and fennel roots to regulate menstruation	[19]		
*Matricaria chamomilla* L. (Chamomile)	Camomilla	Drink a decoction of chamomile flowers to regulate menstruation	[25]		
		Drink fasting a decoction of chamomile and burning bush flowers, early in the morning, to regulate menstruation	[26]		
		Washing with chamomile and poppy flowers, and mallow leaves to treat dysmenorrhea			
		Drink a decoction of chamomile flowers to treat dysmenorrhea	[27]		
		Drink an infusion of Southern maidenhair fern, chamomile, mallow, lettuce, chicory, celery and fennel roots to treat dysmenorrhea	[19]		
		Drink fasting a decoction of parsley root, fennel, mallow leaves and chamomile flowers to regulate menstruation			
		Drink a decoction of chamomile flowers to treat amenorrhea	[28]		
		Drink a decoction of chamomile flowers to treat amenorrhea	[29]		
		Drink a decoction of chamomile flowers to regulate menstruation and to treat dysmenorrhea and amenorrhea	[30]		
		Drink a half dram of powder of chamomile flowers (one once), savin juniper flowers (one once), saffron flowers (two dram), incense seeds (1 crap), pigeon dried blood (three drams), blessed in a half dram of wine, to treat amenorrhea			
*Petasites albus* Gaertn. (White butterbur)	Farfaraccio bianco	Use white butterbur, generically, to treat metrorrhagia	[31]		
*Tussilago farfara* L. (Coltsfoot)	Farfarella	Drink a decoction of coltsfoot and common ivy leaves to regulate menstruation	[24]		
**Boraginaceae**					
*Borago officinalis* L. (Starflower)	Borragine	Eat starflower leaves boiled in the wine to treat amenorrhea	[20]		
**Brassicaceae**					
*Eruca vesicaria* (L.) Cav. (Arugula)	Rucola	To treat, generically, menstrual disorders	[32]		
*Nasturtium officinale* R. Br. (Water Cress)	Crescione d’acqua	Drink a decoction of water cress leaves	[28]		
**Burseraceae**					
*Boswellia* spp. (Incense)	Incenso	Drink a half dram of powder of chamomile flower (one once), savin juniper (one once), saffron (two dram), incense (one crap), pigeon dried blood (three drams), blessed in a half dram of wine, to treat amenorrhea	[30]	Diseases of Women I, 74; Diseases of Women II, 195–197; Nature of Women, 109	[21,22,23]
**Caprifoliaceae**					
*Valeriana officinalis* L. (Valerian)	Valeriana	Eat a soup of valerian leaves and red chickpea to regulate menstruation	[17]		
**Cucurbitaceae**					
*Bryonia dioica* Jacq. (White bryony)	Brionia	Use white bryony root to treat amenorrhea	[18]		
**Cupressaceae**					
*Cupressus sempervirens* L. (Cypress)	Cipresso	Hold on the neck cypress leaves to regulate menstruation	[18]	Diseases of Women I, 86; Diseases of Women II, 192–196;	[21]
		Hold on the skin southern wormwood and cypress leaves to treat amenorrhea			
*Juniperus sabina* L. (Savin juniper)	Erba sabina	Drink a decoction of savin juniper, Southern maidenhair fern leaves and antimony to regulate menstruation	[19]		
		Drink a half dram of powder of chamomile flowers (one once), savin juniper flowers (one once), saffron flowers (two dram), incense seeds (one crap), pigeon dried blood (three drams), blessed in an half dram of wine, to treat amenorrhea	[30]		
**Fabaceae**					
*Acacia senegal* (L.) Willd. (Gum arabic tree)	Gomma arabica	Eat egg whites with Arabic gum to treat metrorrhagia and menorrhagia	[28]	Diseases of Women I, 87	[21]
*Cicer arietinum* L. (Chickpea)	Ceci	Eat a soup of red chickpeas with valerian to regulate menstruation	[17]	Diseases of Women II, 192	[21]
		Eat a soup of chickpeas to regulate menstruation	[19]		
		Eat a soup of red chickpeas to treat amenorrhea	[30]		
*Phaseolus vulgaris* L. (Common bean)	Fagiolo	Eat a soup of red beans to regulate menstruation	[19]		
**Hypericaceae**					
*Hypericum perforatum* L. (St John’s wort)	Erba di San Giovanni	Drink a decoction of St John’s wort to regulate menstruation	[24]		
		Drink a decoction of St John’s wort leaves to treat dysmenorrhea	[33]		
**Iridaceae**					
*Crocus sativus* L. (Saffron)	Zafferano	Drink saffron flowers and white grapes infused in white wine to treat amenorrhea	[22]	Diseases of Women II, 195	[21]
		Hold on the skin saffron flowers to regulate menstruation	[19]		
		Drink a half dram of powder of chamomile flowers (one once), savin juniper flowers (one once), saffron flowers (two dram), incense seeds (one crap), pigeon dried blood (three drams), blessed in a half dram of wine, to treat amenorrhea	[30]		
**Juglandaceae**					
*Juglans regia* L. (Walnut)	Noce	Drink a decoction of walnut leaves to treat menorrhagia	[34]		
		Drink a decoction of walnut leaves to treat menorrhagia	[27]		
**Lamiaceae**					
*Lavandula stoechas* L. *Lavandula angustifolia* Mill. (Spanish lavender, Lavender)	Lavanda	Use, generically, lavender, collected during St. John night, to regulate menstruation	[18]		
*Marrubium vulgare* L. (Common horehound)	Marrubbio	Drink a decoction of common horehound leaves and take a foot bath with hot water and bran early in the morning, and with hot water and cinder in the evening to regulate menstruation	[19]		
*Origanum vulgare* L. (Wild marjoram)	Origano	Eat wild marjoram aerial plants boiled in wine to treat amenorrhea	[20]	Diseases of Women I, 74; Nature of Women, 71	[21,22,23]
*Salvia Rosmarinus* Spenn. (Rosemary)	Rosmarino	Eat, generically, rosemary to treat menstrual disorders	[20]		
*Salvia officinalis* L. (Sage)	Salvia	Drink a decoction of sage leaves to block menstrual flux	[18]	Diseases of Women I, 57, 66; Diseases of Women II, 193	[21]
*Satureja montana* L. (Mountain savory)	Santoreggia	Drink a decoction of mountain savory leaves to treat menstrual disorders	[18]		
*Vitex agnus-castus* L. (Vitex)	Agnocasto	Swallow powder of vitex flowers to regulate menstruation	[35]	Diseases of Women II, 192, 198; Nature of Women, 32	[21,22,23]
		Swallow powder of vitex flowers to treat amenorrhea	[36]		
**Malvaceae**					
*Malva sylvestris* L. (Common mallow)	Malva	Drink an infusion of Southern maidenhair fern, chamomile, mallow, lettuce, chicory, celery and fennel roots to regulate menstruation	[19]	Diseases of Women II, 196	[21]
		Use common mallow generically to regulate the menstruation or to treat amenorrhea	[18]		
		Take a foot bath with common mallow boiled water to treat amenorrhea			
		Drink a decoction of common mallow leaves to treat amenorrhea	[37]		
		Take a foot bath with common mallow boiled water to regulate menstruation or to treat amenorrhea	[38]		
		Drink fasting a decoction of parsley root, fennel, mallow leaves and chamomile flowers to regulate menstruation	[19]		
		Washing with boiled chamomile, and mallow leaves and poppy flowers water	[26]		
**Nymphaeaceae**					
*Nymphaea alba* L. (White water lily)	Ninfea	Hold on the pubis white water lily leaves	[18]		
**Papaveraceae**					
*Papaverum rhoeas* L. (Common poppy)	Papavero	Washing with boiled chamomile, and mallow leaves and poppy flowers water to treat dysmenorrhea	[26]	Diseases of Women II, 113	[21]
**Poaceae**					
*Arundo donax* L. (Giant reed)	Canna	Use, generically, giant reed root to treat amenorrhea	[20]		
		Drink a decoction of giant reed root to regulate menstruation or to treat amenorrhea	[18]		
*Cynodon dactylon* (L.) Pers. (Bermuda grass)	Gramigna	Drink a decoction of Bermuda grass leaves to treat amenorrhea	[37]		
*Triticum* spp. (Wheat)	Grano	Expose the buttock to the vapors of hot water mixed with white flour to regulate menstruation	[22]	Diseases of Women II, 193	[21]
		Drink a decoction of common horehound leaves and take a foot bath with hot water and bran early in the morning, and with hot water and cinder in the evening to regulate menstruation	[19]		
*Zea mays* L. (Maize)	Granturco	Eat red wheat seeds to treat amenorrhea	[30]		
		Drink a decoction of red wheat seeds to treat amenorrhea			
**Pteridaceae**					
*Adiantum capillus-veneris* L. (Southern maidenhair fern)	Capelvenere	Use, generically, Southern maidenhair fern to regulate menstruation	[24]	Diseases of Women II, 192	[21]
		Drink a decoction of savin juniper, Southern maidenhair fern leaves and antimony to regulate menstruation	[19]		
		Drink a decoction of Southern maidenhair fern, chamomile, mallow, lettuce, chicory, celery and fennel roots to regulate menstruation			
		Drink a decoction of Southern maidenhair fern aerial part and take a foot bath with hot water and cinder to regulate menstruation			
		Drink a decoction of Southern maidenhair fern aerial part to treat amenorrhea, dysmenorrhea and to regulate menstruation	[30]		
		Drink a decoction of Southern maidenhair fern aerial part to treat amenorrhea	[17]		
		Hold under shoes Southern maidenhair fern aerial part to treat amenorrhea			
		Drink a decoction of Southern maidenhair fern aerial part to regulate menstruation	[31]		
		Drink a decoction of Southern maidenhair fern aerial part to treat amenorrhea	[28]		
		Drink a decoction of Southern maidenhair fern aerial part collected where the bells are not audible, to treat dysmenorrhea	[29]		
		Drink a decoction of Southern maidenhair fern aerial part to treat amenorrhea			
		Drink a decoction of Southern maidenhair fern aerial part to treat amenorrhea	[37]		
		Drink a decoction of Southern maidenhair fern aerial part to treat amenorrhea	[25]		
**Rosaceae**					
*Fragaria vesca* L. (Wild strawberry)	Fragola	Eat wild strawberry to treat amenorrhea	[18]		
*Prunus dulcis* (Mill.) D.A.Webb (Almond)	Mandorlo	Take a spoon of sweet almond oil early in the morning overnight fasting to treat dysmenorrhea	[26]	Diseases of Women II, 200	[21]
**Rutaceae**					
*Dictamnus albus* L. (Burning bush)	Dittamo	Drink a decoction of burning bush leaves to regulate menstruation	[19]		
		Drink a decoction of burning bush leaves to regulate menstruation or to treat amenorrhea and dysmenorrhea	[30]		
		Drink burning bush and chamomile boiled water early in the morning overnight fasting to treat amenorrhea	[26]		
*Ruta graveolens* L. (Rue)	Ruta	Eat powder rue dried leaves to treat amenorrhea	[28]		
		Use, generically, rue to treat amenorrhea or to regulate menstruation	[18]	Diseases of Women II, 113; Nature of Women, 59	[21,22,23]
**Salicaceae**					
*Salix* spp. (Willows)	Vetrica gentile	Drink a decoction of willow bark in the evening for several days to regulate menstruation	[17]		
**Solanaceae**					
*Solanum nigrum* L. (Black nightshade)	Morella	Drink a decoction of black nightshade leaves to stop menstrual flux	[18]		
**Urticaceae**					
*Parietaria officinalis* L. (Eastern pellitory-of-the wall)	*Erba de spaccapétra*	Use, generically, Eastern pellitory-of-the wall to regulate menstruation	[39]		
*Urtica dioica* L. (Common nettle)	Ortica	Drink a decoction of common nettle leaves to treat metrorrhagia and menorrhagia	[28]	Diseases of Women II, 113	[21]
*Urtica pilulifera* L. (Roman nettle)	*Ardicula masculina*	Take a foot bath with boiled Roman nettle water to treat amenorrhea	[28]	Diseases of Women II, 113	[21]
**Verbenaceae**					
*Verbena officinalis* L. (Vervain)	Verbena	Drink a decoction of vervain to regulate menstruation	[18]		
**Vitaceae**					
*Vitis vinifera* L. (Grapevine)	Vite	Drink a half dram of powder of chamomile flowers (1 once), savin juniper flowers (1 once), saffron flowers (2 dram), incense seeds (1 crap), pigeon dried blood (3 drams), blessed in a half dram of wine, to treat amenorrhea	[30]	Diseases of Women II, 112; Nature of Women, 109	[21,22,23]
		Drink saffron flowers and white grapes infused in white wine to treat amenorrhea	[22]		

**Table 2 plants-13-00589-t002:** Plants used in folk Italian medicine to treat menstrual diseases, between the latter half of the nineteenth century and the early to mid-twentieth century, nowadays employed in human trials to management menstrual diseases.

Family/Scientific /Common Name	Human Trials	Clinical Manifestation PMS, Oligo-, Meno-, Dis-, Metro-, A-menorrhea	Methods and Results	References
**Apiaceae**				
*Foeniculum vulgare* Mill. (Fennel)	Placebo-controlled double-blind clinical study on 50 patients (girls’ mean age: 22); fennel/placebo group to verify the effect of fennel on pain intensity.	Primary dysmenorrhea	Fennel essence was produced by distillation of seeds with water vapor, converted into pearl shapes (capsule 30 mg), and orally administered. Patients divided into two groups were under treatment for two cycles. A capsule of fennel extract, four times a day for 3 days from the first day of the menstrual period was given. In conclusion, fennel is an effective herbal drug for menstrual pain.	[40]
	A semi-experimental single-blind study on 110 girls, mean age 15.5 (55 fennel extract and 55 mefenamic acid), was conducted to compare the effects of fennel and mefenamic acid.	Severe primary dysmenorrhea	Foeniculum extract (Barij Extract Co., Tehran) was orally administered. Patients were categorized into two groups that received either 30 drops of fennel extract or 250 mg of mefenamic acid at the onset of menses and then continuously every 6 h for the first three days. There was no significant difference between the two groups in the level of pain relief.	[41]
	A randomized single-blind, placebo-controlled trial was carried out on 75 single female students who suffered from primary dysmenorrhea. They were divided in three groups receiving fennel extract, vitamin E, and placebo.	Primary dysmenorrhea	Fennel extract was orally administered. For two consecutive menstrual periods, drugs (capsule 46 mg) were used four times a day from the onset of bleeding for a duration of three days. The pain severity decreased in the fennel extract and vitamin E groups. A significant difference was found in pain severity during the second cycle and the reduction was greater in the group receiving the fennel extract.	[42]
	Thirty women (15–24 years) treated for three cycles (control, mefenamic acid, essence of fennel’s fruits)	Primary dysmenorrhea	The essence of fennel’s fruit was obtained through distillation with steam from dried ripe fruits and was orally administered. For the 1st cycle, there were no drugs used, for the 2nd cycle, mefenamic acid (250 mg q6h orally) was used, and for the 3rd cycle, fennel (25 drops q4h orally) was used. Based on the paired t-test analysis, mefenamic acid was more potent than fennel on the second and third days of menstruation. On the other days the difference was not significant.	[43]
	Double-blind trial was carried out in 105 students (18–25 years), randomly divided into four groups which received the extracts of fennelin (N = 25), vitagnus (N = 25), mefenamic acid (N = 30), and placebo (N = 25).	Primary dysmenorrhea	Market herbal drops were orally administered. Treatments were administered one day before the start of the cycle until the third day: fennel group (30 drops every 4 h), vitagnus group (40 drops once a day in the morning), mefenamic acid group (250 mg capsules every 4 h), and placebo group (30 drops every 4 h). Each group was assayed for three cycles, one without any drugs and then two cycles with them. There was no significant difference in the mean of severity of dysmenorrhea during one cycle before the intervention among the four groups, but the difference was significant during the two cycles after the intervention. Fennelin had similar effects as vitagnus on dysmenorrhea. Mefenamic acid had a smaller effect than both of the herbal drugs.	[44]
	Double-blind clinical trial involving 90 randomly selected subjects (46 cases and 44 controls).	Primary dysmenorrhea	Orally administered five capsules containing 46 mg of Foeniculum vulgare and identical placebos were provided to be taken daily by the case and control groups, respectively, during the first three days following the onset of dysmenorrheal pain whenever they needed the medications. Severity of pain in the treated group with Foeniculum vulgare extract in comparison with the placebo group, showed a significant statistical difference.	[45]
	In a single-blind randomized clinical trial, 90 students were randomly divided into three equal groups and they received echinophora-platyloba extract, fennel extracts, and placebo.	Moderate to severe PMS	The extracts were orally administered. There were no significant differences in the means of the premenstrual syndrome scores before the intervention among the three groups, but the differences were significant after the intervention. No significant differences were seen between the echinophora-platyloba and fennel groups.	[46]
*Pimpinella anisum* L. (Anise)	Randomized double-blind controlled clinical trial (84 women, aged 18–35, subdivided into anise and placebo groups)	PMS	Dried and powered seeds were extracted with hydro-alcoholic solvent. In the study group, a capsule of 250 mg with anise extract (110 mg) and maize starch (140 mg), three times a day for ten days from day 21 of the cycle to day 3 of the next cycle was administered; in the placebo group, a capsule of 250 mg containing maize starch was orally administered. The two groups were under treatment for two cycles. Anise was effective in decreasing the symptoms of PMS in comparison to the placebo.	[47]
**Asteraceae**				
*Matricaria chamomilla* L. (Chamomile)	A prospective randomized double-blind trial was performed on 59 women (22.42 ± 2.55) using chamomile and 59 (21.71 ± 2.17) using mefenamic acid (250 mg) to compare the use of chamomile and mefenamic acid on the intensity of PMS.	PMS	Chamomile (flowers) extract was prepared through soaking with ethanol 96% followed by distillation. A capsule containing 100 mg of chamomile extract and 150 mg of starch was orally administered three times a day for two cycles. Chamomile seems to be more effective than mefenamic acid in reducing the intensity of PMS-associated psychological pain symptoms.	[48]
	Clinical trial study: 30 women were evaluated for two cycles on the reduction of dysmenorrhea and PMS utilizing no chamomile in the first cycle and chamomile in the second.	PMS; dysmenorrhea	Thirty drops of chamomile’s flowers’ extract in a glass of water every 8 h was orally administered 3 days before the beginning of menstruation. The greatest effect of chamomile was in the reduction of the severity of anxiety and retention symptoms and thus, chamomile can be used for reducing the severity of the mental and physical symptoms of PMS.	[49]
	A double-blind randomized controlled trial: 30 women used chamomile and 30 used placebos to test the effect on breast pain in premenstrual period.	PMS	Five drops of Chamomile extract, three times a day in a glass of warm water was orally administered, for two consecutive months. Chamomile presents a safe well-tolerated and effective treatment for women with moderate mastalgia.	[50]
	This crossover, triple-blind randomized clinical trial study was performed on 80 students (24.72 ± 2.55 years) with primary dysmenorrhea. They were randomly allocated to mefenamic acid and *Matricaria chamomilla* groups.	Dysmenorrhea	The results of this study showed that taking *Matricaria chamomilla* capsules can decrease the severity of dysmenorrhea, so it is recommended to use it in the treatment of this common gynecologic disorder in women.	[51]
**Boraginaceae**				
*Borago officinalis* L. (Starflower)	A total of 180 female patients (age 30.1 ± 7.43) with a previous clinical diagnosis of PMS.	PMS	One daily 900 mg borage oil capsule, containing 180 mg ɣ-linoleic acid (GLA)/capsule, was orally administered for a minimum of three menstrual cycles; 95.4% of treated patients displayed a reduction in PMS symptoms. The use of *Borago officinalis* oil was safe and effective in the treatment of the physical and emotional symptoms of PMS in the patient population evaluated.	[52]
**Burseraceae**				
*Boswellia* spp. (Incense)	Double-blind randomized, placebo-controlled clinical trial was conducted in two gynecology outpatient clinics in Isfahan city (Iran) from August 2016 to January 2018. Patients with heavy menstrual bleeding (n = 102; age from 18 to 45 years) were randomly assigned to three groups.	Menorrhagia	All patients received ibuprofen (200 mg) and either frankincense (300 mg produced from oleoresin from trunk and twigs, containing Boswellic acid derivated), ginger (300 mg), or a placebo, which contained 200 mg of anhydrous lactose as the filling agent and was similar in appearance to the two other drugs. Patients received the medications three times a day for seven days of the menstrual cycle, starting from the first bleeding day and this was repeated for two consecutive menstrual cycles. Ginger and frankincense seem to be effective complementary treatments for heavy menstrual bleeding. Further studies with a larger sample size and longer follow-up are warranted in this regard.	[53]
**Caprifoliaceae**				
*Valeriana officinalis* L. (Valerian)	In a double-blind, randomized, placebo-controlled trial, 100 students were randomly assigned to receive valerian (n = 51 mean age 20.90) or a placebo (n = 49 mean age 21.04).	Dysmenorrhea	Valerian (dose 255 mg obtained from roots and rhizome) was orally administered three times daily for 3 days beginning at the onset of menstruation, for two consecutive menstrual cycles. Valerian seems to be an effective treatment for dysmenorrhea, probably because of its antispasmodic effects.	[54]
	A double-blind clinical trial with 100 females; placebo group mean age, 22.55; and valerian group, 22.48.	PMS	Oral administration of two pills daily in the last seven days of their menstrual cycle for three cycles. The results suggest that the extract could reduce the severity of symptoms of premenstrual syndrome.	[55]
**Hypericaceae**				
*Hypericum perforatum* L. (St John’s wort)	Pilot study involving 19 women aged between 19–50 years old.	PMS	Hypericum tablets for two complete menstrual cycles (1 × 300 mg hypericum extract per day standardized to 900 µg hypericin) were orally administered. There were significant reductions in all outcome measures. The degree of improvement in the overall premenstrual syndrome scores between baseline and the end of the trial was 51%, with over two-thirds of the sample demonstrating at least a 50% decrease in symptom severity. Tolerance and compliance with the treatment were encouraging. The results of this pilot study provide a basis for the hypothesis that hypericum is a useful treatment for premenstrual syndrome. This should now be investigated in a randomized, placebo-controlled, double-blind trial.	[56]
	The participants were randomly divided into two groups. Mean age was 31.23.	PMS	Two 680 μg hypericin tablets per day and a placebo (two cellulose tablets per day) were orally administered for 8 weeks (two menstrual cycles). H. perforatum, which is cheap and commonly available, is a well-tolerated and effective drug for the treatment of women with moderate–severe PMS.	[57]
	Randomized, double-blind trial placebo-controlled study.	PMS	The hypericum or placebo, 30 drops BD, was orally administered for two complete cycles. Premenstrual daily hypericum caused a significant improvement in the mean daily score of severity of premenstrual problems compared to the placebo.	[58]
	Randomized, double-blind, placebo-controlled study, involving 36 women aged 18–45 years.	PMS	*Hypericum perforatum* tablets, 900 mg/day (standardized to 0.18% hypericin; 3.38% hyperforin), or identical placebo tablets were administered for two menstrual cycles. After a placebo-treated washout cycle, the women crossed over to receive the placebo or *Hypericum perforatum* for two additional cycles. Daily treatment with *Hypericum perforatum* was more effective than the placebo treatment for the most common physical and behavioral symptoms associated with PMS.	[59]
**Iridaceae**				
*Crocus sativus* L. (Saffron)	The study was a double-blind, placebo-controlled study involving short-term exposure (20 min) to stimuli. Forty-seven women were randomly assigned to a saffron group (n = 36; 18 in follicular phase, 18 in luteal phase) or a control group (n = 11; 5 in follicular phase, 6 in luteal phase). The study evaluates hormone levels in saliva (after inhalation) and investigates the psychological effect of saffron by STAI test.	PMS, dysmenorrhea, irregular menstruation	Administration was through the smelling of ethylalcohol extract of saffron flowers that contain saffranal, for 20 min a day. The present findings support the existence of physiological and psychological effects of saffron odor in women. Our results indicate that saffron odor exerts some effects on the treatment of PMS, dysmenorrhea and irregular menstruation. This is the first report to suggest that saffron odor may be effective in treating menstrual distress.	[60]
	Double-blind, randomized and placebo-controlled trial. 47 women aged 20–45 years were randomly assigned to receive capsule saffron or capsule placebo (24 saffron vs. 23 placebo).	PMS	Capsule saffron (dried extract of petals, 30 mg/day: 15 mg twice a day; morning and evening) was orally administrated for two menstrual cycles. The results of this study indicate the efficacy of *C. sativus* L. in the treatment of PMS. However, a tolerable adverse effects profile of saffron may confirm the application of saffron as an alternative treatment for PMS. These results deserve further investigations.	[61]
	Randomized triple-blind and placebo-controlled trial: 78 women (39 saffron vs. 39 placebo) aged 18–35 years.	PMS	A 30 mg capsule of dried ethyl alcohol extract of saffron stigma was orally administered once a day for two menstrual cycles. The results of this study suggest that saffron reduces the severity of PMS symptoms, but in order to prove its effectiveness for the treatment of this syndrome, further research is warranted.	[62]
	Double-blind randomized placebo and mefenamic acid trial. One hundred and eighty participants in total: group 1 (60): received mefenamic acid 250 mg capsules; group 2 (60): saffron 30 mg capsules (in 100 mg Saffron powder); group 3 (60): placebo capsules.	Dysmenorrhea	Three capsules (SAHARKHIZ saffron) were orally administered per day for three days during three consecutive menstrual cycles. It was found that the effect of saffron in reducing pain is more than mefenamic acid and far more than the placebo.	[63]
**Lamiaceae**				
*Lavandula stoechas* L. *Lavandula angustifolia* Mill. (Spanish lavender, Lavender)	A randomized controlled trial among 102 students randomly divided into massage, isometric exercises, and control group (34 in each experimental group and 34 in the control group). Age: control group, 21.08; massage group, 21.41; exercise group, 20.73. The exercise group received two consecutive cycles of effleurage massages with lavender oil. The second group had 8 weeks of isometric exercises. No intervention was performed by the control group.	Primary dysmenorrhea	Lavender extract based on olive oil with 10% purity was used. With the patients in a supine position, some lavender oil was applied on the massage spot and effleurage massaging of the upper part of the symphysis pubis and umbilicus was started in a clockwise manner (each for 15 min) at the peak of menstrual pain (usually on the first day). On the second day of menstruation, all the steps of the previous day were equally repeated for all the participants at almost the same hour of the day. Pain intensity had significantly reduced in the massage and exercises groups; the reduction was more significant in the massage group. Based on the present findings, it seems that massage therapy and isometric exercises are effective in reducing some symptoms of dysmenorrhea.	[64]
	This study, a triple-blind randomized clinical trial, was carried out on 200 female students (19–29-year-olds). Subjects were divided into two lavender (A) and diluted milk (B) groups.	Primary dysmenorrhea	Groups A and B were offered a glass of 10 cc lavender essence and 10 cc diluted milk, respectively. Then, the subjects were asked to strew three drops of the solution onto a piece of cotton and smell that for the first 3 days of menstruation, once daily for 30 min in two continual cycles. There was a significant difference in the average pain severity between the treatment and control groups after intervention. Using lavender aromatherapy for 2 months may be effective in decreasing the pain severity of primary dysmenorrhea. The findings showed that lavender had a significant effect on primary dysmenorrhea.	[65]
	A quasi-experimental design study with the subjects as their own control. Forty-four female students (age 20.31) that receive an aromatherapy massage (lavender oil) and a placebo.	Dysmenorrhea	Three menstrual period documentation of pain level during the administration of an aromatherapy massage or placebo; a participant who had previously received the placebo received aromatherapy and in order to compare, the participant who received aromatherapy received the placebo. The massage was carried out for 15 min at a fixed time of day. Two ml of lavender oil or placebo was used and the massage was applied on the abdomen. The results of the study have demonstrated that massages are effective in reducing dysmenorrhea. In addition, this study also demonstrated that the effect of aromatherapy massages on pain was greater than that of placebo massages.	[66]
	Randomized crossover study to evaluate the efficacy of lavender oil on mood states and autonomic nervous system activity during PMS. Seventeen women (20.6) were examined on two separate occasions (aroma or control trials) in the late luteal phase (within seven days before the next menstruation).	PMS	Aroma lavender or water stimulation were administered by inhalation for 10 min. Ten µL of lavender oil or water was used for inhalation. The study indicated that short-term inhalation of lavender could alleviate premenstrual emotional symptoms and could, at least in part, contribute to the improvement of parasympathetic nervous system activity.	[67]
	Randomized clinical trial. Ninety-six female students (18–28 years old) divided into two groups: n = 48, who inhaled lavender-based sesame oil and n = 48, who inhaled sesame oil only.	Primary dysmenorrhea	The students were asked to strew three drops of the treatment (lavender or placebo) on their palms, rub them together, keep their hands at a distance of 7–10 cm from their nose and inhale for 5 min. The treatments were administered for 1 h after experiencing dysmenorrhea and every 6 h for the first three days of menstruation. During the two consecutive menstrual cycles, one of two treatments was administered to the students. The results did not show a significant effect of lavender on menstrual bleeding. Moreover, this study showed that lavender inhalation was effective in alleviating dysmenorrhea symptoms, suggesting that it could be applied by midwives in a safe manner because it has no side effects, and the simplicity and cost-effectiveness for all patients.	[68]
	This study was a clinical trial. Eighty students (18–24 years) were involved. Each participant, in the first few days of menstruation, randomly received two types of massages with either lavender or placebo oil during two consecutive cycles of menstruation.	Primary dysmenorrhea	Two drops of lavender essential oil were mixed with 5 mL of sweet almond oil and a 2% concentration of the oil was prepared. The intervention was performed at the onset of pain coinciding with the onset of menstruation. The oil used in this study was prepared from blooming branches of lavender. The researcher poured 2 mL of lavender oil or placebo on their hands and massaged with rotation movements using both hands without creating pressure on the abdomen for 15 min, while the subjects lay on their backs. Lavender oil, by reducing anxiety, causes more pain relief. Findings of this study showed that lavender oil massages decrease primary dysmenorrhea and it can be used as an effective herbal drug.	[69]
	A randomized controlled trial design: 40 women with inhalation aromatherapy using lavender oil and 37 women in the control group. The intervention and control groups were followed up for three cycles.	PMS	To administer the aromatherapy, 3,15 mL of lavender oil was required. Three drops of lavender oil were added into 200 mL of hot water; the participant’s head was covered with a towel in the sitting position and then she inhaled the steam. The participants’ group were also instructed to begin the aromatherapy at least 10 days before their cycle, and to complete it once a day at the same hour, and to stop it when their periods started. It was concluded that inhalation aromatherapy can be used for coping with PMS. PMS symptoms such as anxiety, depressive affect, nervousness, pain, bloating, and depressive thoughts are decreased by inhalation aromatherapy, but it has no effect on appetite and sleep changes.	[70]
	A randomized placebo-controlled trial: 67 women, mean 20.7 years.	PMS	Abdominal massage was performed in three groups: experimental group (25 women) was treated with two drops of lavender, one drop of clary sage and one drop of rose in 5 cc of almond oil; placebo group (20 women) was treated with almond oil; and control group (20 women) was not treated. The experimental group showed significant improvement in dysmenorrhea as assessed by the verbal multidimensional scoring system.	[71]
	A randomized, double-blind clinical trial: 48 women, mean 24.5 years.	Primary dysmenorrhea	Abdominal massage was performed in two groups: experimental group (24 women) was treated with two drops of lavender, one drop of clary sage and one drop of Marjoram added into a 3% massage cream in an unscented jojoba cream; the placebo group (24 women) was treated with synthetic fragrance added to the jojoba cream. Aromatherapy massage is more effective when using an essential oil than a synthetic fragrance in terms of relieving primary dysmenorrhea.	[72]
*Salvia Rosmarinus* Spenn. (Rosemary)	Randomized double-blind study was conducted on 82 students with primary dysmenorrhea (female aged 18–25 years). Group 1 (40): received mefenamic acid; group 2 (42): received rosemary.	Dysmenorrhea	Participants in the mefenamic acid group and rosemary group received 250 mg capsules from the onset of the menstrual period, every 8 h, for two cycles in the first three days of menstruation. Rosemary capsules reduce menstrual bleeding and primary dysmenorrhea the same as mefenamic acid capsules.	[73]
*Salvia officinalis* L. (Sage)	A triple-blind randomized clinical trial on 90 college students. The inclusion criteria were diagnosis of PMS, regular menstrual cycle, aged 18–35 years, normal body mass index (BMI), and no physical or psychological illness conditions.	PMS	The participants were randomly divided into two groups who were treated with 500 mg *Salvia officinalis* capsules or a placebo once a day for two consecutive months. The intervention and placebo group members each received one capsule of *Salvia officinalis* or placebo, respectively, from the 21st day of menstruation till the 4th day of the next cycle for two consecutive menstrual cycles. The between-group comparison showed a significantly higher reduction in the severity scores for *Salvia officinalis* compared to the placebo, except for ‘anxiety’, ‘suicide tendency’, and ‘increase or loss of appetite’, indicating the higher effectiveness of *Salvia officinalis* in alleviating the PMS physical symptoms. In conclusion, *Salvia officinalis* is an effective alternative agent to reduce the severity of psychological and physical symptoms of PMS.	[74]
*Vitex agnus-castus* L. (Vitex)	Randomized, double-blind, placebo-controlled, parallel-group comparison over three menstrual cycles. A total of 170 women (86 agn, 84 placebo); mean age was 36 years.	PMS	Agnus castus (dry extract from fruits tablets: extract ZE 440: 60% ethanol m/m, extract ratio 6–12:1; standardized for casticin) in the form of one 20 mg tablet or a matching placebo was administered once daily orally for three consecutive cycles. The plant has been used traditionally to relieve the symptoms of the premenstrual syndrome, although systematic evaluation of its efficacy is relatively recent. The dry extract of agnus castus fruit is an effective and well-tolerated treatment for the relief of premenstrual syndrome symptoms.	[75]
	A large, prospective and randomized study over three consecutive menstrual cycles in adolescents aged 17–25 years suffering from moderate-to-severe premenstrual syndrome (PMS). Seventy-two patients were enrolled in the study and randomly assigned to receive *Vitex agnus-castus* (27), or contraceptive pills (25) or placebo once a day (20).	PMS	For three consecutive cycles, one tablet of *Vitex agnus-castus*, or new oral contraceptive pills (containing 3 mg drospirenone and 30 μg di etinilestradiolo, in the 24/4 formulation) or a placebo was administered once a day. There was a significant difference in the PMSD sum score in the three groups, while the efficacy rate in the treatment groups was significantly higher than the placebo group, the results from the groups treated with VAC and treated with OC are comparable. The tolerability of agnus-castus was good in our study and higher than that of the oral contraceptive. In conclusion, *Vitex agnus-castus* is a safe, well-tolerated and effective drug for the treatment of moderate-to-severe PMS in adolescents.	[76]
	The randomized and controlled study was to evaluate the efficacy and tolerability of a fast-acting form of agnus castus fruit extract (Monoselect Agnus, MA) administered for 90 days, in comparison with a magnesium supplement for women with premenstrual syndrome (PMS). Eighty-two women were enrolled [MA: 42 (age 37 ± 9.1); magnesium: 40 (age 36 ± 8.2)].	PMS	Extract obtained by extraction with 60% ethanol from ripe fruits of *Vitex agnus castus*, titrated to 0,5% in agnuside (extract provided by Indena, Milan, Italy). MA (40 mg/tablet) or matching Magnesium (300 mg/tablet) was administered in the form of one tablet daily for the first three consecutive months. For the following 3 months, only MA was administered (one tab/day for 7 days/month). The results obtained demonstrate that MA is an effective and well-tolerated treatment for the relief of symptoms of premenstrual syndrome. Its action is clearly evident after 90 days of continuous treatment and a 7 days/month treatment scheme can be maintained.	[77]
	A prospective, double-blind, placebo-controlled, parallel-group, multicenter clinical trial design was employed in this study. Two groups (aged 18–45): treatment group (101); placebo group (101).		A tablet of VAC BNO 1095 contained 4.0 mg of dried ethanolic (70%) extract of VAC (corresponding to 40 mg of herbal drug). The VAC BNO 1095 extract was administered orally once daily throughout the three cycles during the treatment phase (TP) to the subjects of the treatment group. *Vitex agnus castus* is a safe, well-tolerated and effective drug treatment for patients with moderate-to-severe premenstrual syndrome, and the effects are confirmed by physicians and patients alike.	[78]
	Multicenter, controlled, randomized, comparative trial versus pyridoxine (18–45 years). *Vitex agnus castus* (VAC): n = 46; pyridoxine (B6).		Patients in the VAC group received one capsule of the Agnolyt© (1 capsule contains: dried extract of the chaste tree fruit (3.5–4.2 mg) and one capsule of the placebo daily. Patients in the pyridoxine group received one capsule of the placebo twice daily on days 1 to 15, and one capsule of pyridoxine-H CL (100 mg) twice daily on days 16 to 35 of the menstrual cycle. At first glance, the effect of pyridoxine appears to be equivalent to that of VAC in the present comparative trial; however, careful evaluation of all available data and scales permits the conclusion that VAC is superior to pyridoxine in the present study.	[79]
	A multicenter, double-blind, placebo-controlled, randomized, prospective, parallel-group study, to test different doses of Ze440 (8, 20 e 30 mg) over three menstrual cycles; 142 patients divided into 35 (placebo), 36 (8 mg agnus castus), 35 (20 mg agnus castus) and 36 (30 mg agnus castus).		One tablet should be taken once daily unchewed with some liquid during a meal for the entire three-menstrual cycle period (20 mg native 60% ethanolic extract m/m corresponding to 180 mg VAC crude drug per day). This study demonstrated that the VAC extract, Ze 440, was effective in relieving symptoms of PMS, when applied at a dose of 20 mg once daily.	[80]
	Randomized, placebo-controlled, double-blind, cross-over study with 128 women: treatment n = 62 (age 30,77); placebo n = 66 (age 30,89).	PMS	Forty drops of Vitex agnus extract or matching placebo, orally administrated for 6 days before menses for six consecutive cycles. Vitex agnus can be considered as an effective and well-tolerated treatment for the relief of symptoms of mild and moderate PMS.	[81]
	A double-blind, randomized, controlled trial on a volunteer sample: 105 VAC and 112 soya-based placebos (18–46 years old).	PMS	Three-hundred mg tablets of powdered Vitex (two tablets to be orally taken three times per day) tested against a soya-based placebo for three months. The results of this clinical trial indicate that Vitex has very little difference in its treatment effect to that of the placebo substance in the majority of symptoms strongly associated with PMS.	[82]
	A prospective, multicenter trial to test the efficacy of *Vitex agnus castus* L extract Ze 440 (V 23/95, 60% ethanol m/m, drug extract ratio: 6–12:1; Zeller AG, CH-Romanshorn) in 43 patients (age 31.3 ± 7.7 years).	PMS	The patients were treated daily with one tablet (20 mg native extract) orally administered during three menstrual cycles. The dose was 1 × 1 tablet daily for three menstrual cycles. In conclusion, patients with PMS can be treated successfully with *Vitex agnus-castus* extract Ze 440, as indicated by a clear improvement in the main effect parameter during treatment and the gradual return after cessation of treatment. The main response to treatment seems related to symptomatic relief rather than the duration of the syndrome.	[83]
	A multicenter, prospective, open-label, single-arm, phase 3 study was performed in 67 Japanese women with PMS and aged 18–44 years.	PMS	The study drug Prefemin containing 20 mg of VAC extract as an active ingredient, was manufactured, and provided by Max Zeller Söhne AG, Romanshorn, Switzerland. VAC extract is a 60% m/m aqueous–ethanolic extract from the fruit of the chaste tree, with a DER of 6–12:1. Patients orally received one tablet daily for three menstrual cycles. The VAC extract improved PMS symptoms in Japanese patients, with no substantial adverse events.	[84]
	A multicentric noninterventional trial (open study without control) in 954 patients (35.8 years).	PMS	Femicur^®^ capsules (Schaper & Brümmer GmbH & Co. KG, Salzgitter, Germany) (one capsule containing 1.6–3.0 mg dried ethanolic extract (60%) of Agni casti fructus [6.7–12.5:1]) correspond to 20 mg of the drug. One capsule twice a day was orally administered for three menstrual cycles. Vitex was proven to be effective with respect to all psychic and somatic symptoms of heterogeneous and multifaceted PMS.	[85]
	Prospective, randomized, placebo-controlled, double-blind study in 67 women with fertility disorders: 37 with oligomenorrhea (17 verum, 20 placebo) and 30 with amenorrhea (16 verum, 14 placebo), aged 18–40 years.	Oligomenorrhea and amenorrhea	Fifty drops of Phyto Hypophyson L or placebo was orally administered three times a day over 3 months or three cycles. In women with sterility and oligomenorrhea, a treatment with Phyto Hypophyson L can be recommended over a period of 3 ± 6 months. While especially in women with amenorrhea, the complex agent showed no significant effect.	[86]
	Prospective, randomized, placebo-controlled, double-blind study in 67 women (33 VAC + 34 placebo), 18–45 age	Mild-to-severe PMS	One tablet daily orally administered for three cycles. Tablet contains 4.0 mg of dried ethanolic (70%) extract of VAC. VAC treatment was more effective than the placebo in women with mild-to-severe PMS, especially with symptoms of negative effect and insomnia. Pain severity decreased only in the wheat germ extract group and there was no statistically significant change in the placebo group. Consumption of wheat germ extract mitigated the severity of the general, physical, and psychological symptoms, which was observed after the first month of treatment.	[87]
**Poaceae**				
*Triticum* spp. (Wheat)	Triple-blind randomized clinical trial on 80 women (42 using wheat germ extract and 38 using placebo) with a mean age of 20–45 years.	Dysmenorrhea	Three 400 mg capsules of wheat germ extract or placebo were orally administered daily, between the 16th day of the menstrual cycle and the fifth day of the next menstrual cycle for two consecutive months. Wheat germ extract seems to be an effective treatment for dysmenorrhea and its systemic symptoms, probably because of its anti-inflammatory effects.	[88]
	A triple-blind clinical trial on 84 women divided into two groups (wheat germ extract and placebo) with a mean age of 20–45 years.	PMS	Three 400 mg capsules of wheat germ extract or placebo were orally administered daily٫ between the 16th day of the menstrual cycle and the fifth day of the next menstrual cycle for two consecutive months. The results showed that wheat germ extract significantly reduced the severity of general, physical, and psychological symptoms of PMS.	[89]
**Salicaceae**				
*Salix* spp. (Willows)	A double-blind crossover randomized clinical trial involving 96 female students with level two or three primary dysmenorrhea: 48 students in the treatment group (sequence I) followed by control (sequence II) and 48 students in the control group (sequence I) followed by treatment (sequence II). Mean age was 20.73.	Dysmenorrhea	The intervention was a salix capsule containing ethanolic extract 70% with salicin (400 mg daily) and the active control was a mefenamic acid capsule (750 mg daily) orally administered. During the first two cycles, the severity of dysmenorrhea was measured without any intervention. The students were randomly allocated into two groups of A and B, using a randomized block design of size four in the two next consecutive cycles. In cycle 3, group A received salix capsules as the intervention and group B received mefenamic acid capsules as the active control. Reversely, in cycle 4, group A received mefenamic acid capsules and group B received salix capsules. The duration between two consecutive menstrual cycles was considered as a washout period. Note that for mefenamic acid, the half-life is approximately 2 h and salicylic acid delivered from willow bark has a half-life of approximately 2.5 h. Salix extract decreases dysmenorrhea significantly in the students with primary dysmenorrhea compared with the mefenamic acid and had no adverse effect.	[90]

**Table 3 plants-13-00589-t003:** Plants used in folk Italian medicine to treat menstrual diseases, between the latter half of the nineteenth century and the early to mid-twentieth century, nowadays employed in non-conventional medicine to manage menstrual diseases.

Family/Scientific/Common Name	Diseases/Conditions	Mode of Administration/Part Plant Used	References
**Adoxaceae**			
*Sambucus nigra* L. (Elderberry)	Amenorrhea	Use, generically, elderberry bark	[91]
**Apiaceae**			
*Apium graveolens* L. (Celery)	Menstrual disorders	A leaf is boiled in 500 mL of water and it is drunk periodically, until symptoms disappear	[92]
*Petroselinum crispum* (Mill.) Fuss (Parsley)	Menstrual cramps	Drink an infusion of the whole parsley plant	[93]
	Regulate menses	One teaspoon of parsley fresh roots in one cup of water taken orally	[94]
	Amenorrhea, regulating menstruation	Drink a decoction of parsley aerial parts	[91]
**Araliaceae**			
*Hedera helix* L. (Common Ivy)	Dysmenorrhea	Drink an infusion of common ivy leaves	[95]
	Dysmenorrhea	Drink a decoction of common ivy leaves	[96]
**Asteraceae**			
*Artemisia absinthium* L. *Artemisia pontica* L. (Wormwood)	Regulate menstruation	Drink an infusion of wormwood leaves	[93]
	Dysmenorrhea, irregular menstruation, womb problems	One teaspoon of wormwood fresh leaves and roots in one cup of water taken orally	[94]
	Amenorrhea, dysmenorrhea	Drink an infusion/decoction of wormwood leaves/flowers	[97]
*Cichorium intybus* L. (Blue daisy)	Regulating menstruation	Drink a decoction of blue daisy leaves and roots	[98]
	Stimulate menstruation	Drink a decoction of blue daisy	[99]
*Lactuca sativa* L. (Lettuce)	Painful menses	Drink an infusion of lettuce leaves	[94]
*Tussilago farfara* L. (Coltsfoot)	Menses problems	Drink an infusion of coltsfoot leaves and flowers	[100]
**Cupressaceae**			
*Cupressus sempervirens* L. (Cypress)	Dysmenorrhea	Extracts of common barberry bark or roots with cupressus pseudo-fruits are used to prepare syrups. Take two spoonsful at intervals of 2 h.	[101]
**Fabaceae**			
*Cicer arietinum* L. (Chickpea)	Regulate menstruation	Chickpea seeds are roasted and pounded and given within 15 days after the first menstrual cycle	[102]
**Jungladaceae**			
*Juglans regia* L. (Walnut)	Amenorrhea	Decoction of fruit rind along with seeds of *Raphanus sativus* and jaggery are taken orally till menstruation starts.	[103]
	Amenorrhea	Drink a decoction of walnut leaves	[91]
**Lamiaceae**			
*Marrubium vulgare* L. (Common horehound)	Menstrual pains	Drink an infusion of common horehound aerial parts	[104]
	Regulate menstruation	Use macerate flowers	[105]
	Dysmenorrhea	Drink a decoction of common horehound roots	[106]
	Amenorrhea	Not specified	[107]
*Origanum vulgare* L. (Wild marjoram)	Menstrual disorders	A handful of the fresh plant is boiled in 1 l of water. It is sweetened with honey and left to cool. It is drunk as a soft drink during the day at one’s own discretion. It can also be used to season soups and stews, with the same therapeutic results.	[92]
	Dysmenorrhea	Drink a decoction of wild marjoram leaves	[91]
**Malvaceae**			
*Malva sylvestris* L. (Common Mallow)	Regulate menstruation	Drink an infusion or syrup of common mallow leaves	[93]
	Gynecological diseases	Drink an infusion and decoction of common mallow leaves and aerial parts	[98]
	Dysmenorrhea	Drink a decoction of common mallow leaves and roots	[91]
**Poaceae**			
*Arundo donax* L. (Giant reed)	Dysmenorrhea	Use, generically, stem, leaf, and fruit of giant reed	[95]
	Amenorrhea, dysmenorrhea	Use, orally, giant reed leaves and roots	[91]
*Cynodon dactylon* (L.) Pers. (Bermuda grass)	Amenorrhea	Fresh plant parts are ground and mixed in rice soup and taken	[102]
	Hypomenorrhea	Fresh whole plant is ground and made into the form of a paste. Four teaspoons of the paste are mixed into half a glass of water and consumed in the morning on an empty stomach for 3 days.	[102]
	Amenorrhea	Fresh whole plant is ground and mixed in rice soap and consumed	[108]
	Oligomenorrhea, amenorrhea	Powder of the whole plant is mixed with hot water to drink	[108]
	Menorrhagia/metrorrhagia	Oral administration of the juice of the plant with honey two to three times a day for a few days to treats menorrhagia. Local application of the plant extract on the abdomen in the form of a paste to reduce bleeding in the vagina.	[109]
	Menorrhagia/metrorrhagia	The grass is pounded and filtered to obtain the juice. Half a cup of juice with sugar is taken daily for a week to stop excessive bleeding during menstruation.	[110]
	Menorrhagia	Whole plant juice is given to control heavy menstrual flow and used in the pregnancy stage for strengthening the uterus.	[111]
	Uterine and menstrual pains	Drink a decoction of the aerial parts/rhizome of the Bermuda grass	[104]
	Dysmenorrhea	Drink a decoction of the whole plant/rhizome Bermuda grass	[91]
**Pteridaceae**			
*Adiantum capillus-veneris* L. (Southern maidenhair fern)	Amenorrhea, dysmenorrhea, regulating menstrual cycle	Drink a decoction of the aerial part of the Southern maidenhair fern	[91]
**Rosaceae**			
*Fragaria vesca* L. (Wild strawberry)	Dysmenorrhea	Infusion of wild strawberry leaves	[95]
	Amenorrhea	Drink a decoction of wild strawberry roots	[91]
**Rutaceae**			
*Ruta graveolens* L. (Rue)	Regulate menstruation	Infusion of rue leaves and aerial parts	[93]
	Regulate menses	Not specified	[94]
	Menstrual colic	Drink an infusion of rue leaves	[112]
	Stopping and down menstruation	Drink an infusion of rue leaves	[95]
	Amenorrhea and dysmenorrhea	In small doses, drink a decoction of rue leaves	[107]
**Urticaceae**			
*Parietaria officinalis* L. (Eastern pellitory of the wall)	Dysmenorrhea	Not specified	[113]
*Urtica dioica* L. (Common nettle)	Menstrual complication	Use, generically, common nettle leaves	[99]
	Heavy menstrual bleeding	Drink an infusion of common nettle leaves	[114]
	Amenorrhea, Dysmenorrhea, Menorrhagia	Use, generically, common nettle raw aerial parts	[91]
**Verbenaceae**			
*Verbena officinalis* L. (Vervain)	Regulate menstruation	Drink an infusion of vervain leaves and stems	[115]

## Data Availability

Not applicable.

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
