# Peer review of "Plants in Menstrual Diseases: A Systematic Study from Italian Folk Medicine on Current Approaches"

_plants, 2024, doi:10.3390/plants13050589_

Round 1
Reviewer 1 Report
Comments and Suggestions for Authors
A brief summary
The manuscript “Plants in menstrual diseases: a systematic study from Italian folk medicine to current approaches” reported the use of Plants in menstrual diseases.
The references are recent,but they should include DOI.
Moreover, the references are not correctly reported according to the journal guidelines.
General concept comments
Why is it defined “systematic” review?
The manuscript is review but it is reported as article at line 1.
Specific comments
Lines 267-271. Rosmarinus officinalis name’s is currently Salvia rosmarinus. Moreover, carnosic acid is well known also for its antimicrobial activity. Recently it as been investigated for its antibiofilm activity on S.aureus. This part should include several references according to the different activity. The same for Salvia spp. Lines 272-282.
Comments on the Quality of English LanguageMinor editing of English language required
Author Response
Authors thanks reviewer for his suggestions and comments.
All the references are reported according to the journal guidelines. Moreover DOIs, where available are added.
General concept comments
Why is it defined “systematic” review?
The manuscript is review but it is reported as article at line 1.
In accord to review’s suggestions “systematic” is erased
The type of paper will be changed in “review” by the editor
Specific comments
Lines 267-271. Rosmarinus officinalis name’s is currently Salvia rosmarinus. Moreover, carnosic acid is well known also for its antimicrobial activity. Recently it as been investigated for its antibiofilm activity on S.aureus. This part should include several references according to the different activity. The same for Salvia spp. Lines 272-282.
In accord to review’s suggestion Rosmarinus officinalis L. has been changed in Salvia rosmarinus Spenn.
The activity of carnosic acid reported in the discussion is referred only to its activity in treating MD.
Comments on the Quality of English Language
Minor editing of English language required
Minor editing of English language is done
Reviewer 2 Report
Comments and Suggestions for Authors
The manuscript presents the results of research about plants used in menstrual problems reported in authentic Italian books. This approach is good and reduces the effect of globalization and information exchange nowadays.
The authors provide a very rich list of plants used to treat menstrual disorders and information about the methods of application This information is well systematized. Also, the overall result is interesting. The authors found the proportion of 66.7% of plants utilized in Italian folk medicine from the latter half of the nineteenth to the early mid-twentieth century to treat MD known and utilized nowadays for the same purpose.
In fact, 25.9% of plants exhibit effective properties in human 160 trials for the management of menstrual diseases. (Table 2). On the other hand, 40.8% of plants 161 have been widely used by women globally, as non-conventional medicine (Table 3).
On the other hand, 33.3% of the plants are not observed in the present scientific literature for the management of MD.
In my opinion, these 33% deserve more attention. Are these only the plants used for rituals and magical practices - in that case here is needed a sentence to explain so? If there are more plants in this group some more discussion on their pharmacological effects and even why not a Table 4. The special accent of all this should be on which of them might be prospective candidates for further research on their properties to treat menstrual problems.
Also, some of the plants listed in the tables are toxic in a dose-dependent manner due to their content of various alkaloids e.g. Solanum nigrum, members of Boraginaceae, or other toxic compounds. A section discussing the toxicity would be a good addition.
Tables need their captions
"Compositae" is a synonym of Asteraceae, please use the accepted name and keep once the synonym in brackets when the family is mentioned for the first time. The same is valid for "Leguminosae" - should be Fabaceae
Minor grammar editing of English is required
for example
In fact, 25.9% of plants exhibits effective properties in human 160 trials for the management of menstrual diseases. (Table 2). On the other hand, 40.8% of plants have been widely used by women globally, as non-conventional medicine (Table 3).
In Table 3 some plant names need italicization formatting. Also, the common names need formatting either italic or normal.
Comments on the Quality of English LanguageMinor grammar editing of English is required
for example
In fact, 25.9% of plants exhibits effective properties in human 160 trials for the management of menstrual diseases. (Table 2). On the other hand, 40.8% of plants have been widely used by women globally, as non-conventional medicine (Table 3).
Author Response
Authors thanks reviewer for his useful suggestions and comments.
In particular:
"In my opinion, these 33% deserve more attention. Are these only the plants used for rituals and magical practices - in that case here is needed a sentence to explain so? If there are more plants in this group some more discussion on their pharmacological effects and even why not a Table 4"
Answer:
All the plants used for rituals and magical practices are reported in the paragraph “Rituals and magical practices”
Moreover, as reported in the paragraph “Botanical Analysis and Therapeutic Applications” 33.3% of the plants are not observed in the present scientific literature for the management of MD or in surveys about non-conventional medicine.
"Also, some of the plants listed in the tables are toxic in a dose-dependent manner due to their content of various alkaloids e.g. Solanum nigrum, members of Boraginaceae, or other toxic compounds. A section discussing the toxicity would be a good addition."
Answer: We share the reviewer's reflection but believe that a paragraph on dose-dependent toxicity deviates from the focus of our paper
"Tables need their captions"
Answer: We apologize for the oversights in the now provided table captions.
""Compositae" is a synonym of Asteraceae, please use the accepted name and keep once the synonym in brackets when the family is mentioned for the first time. The same is valid for "Leguminosae" - should be Fabaceae"
Answer: We apologize for the other oversights and have provided substitutions, replacing Compositae with Asteraceae and Leguminosae with Fabaceae.
"Minor grammar editing of English is required"
Answer: Done
Reviewer 3 Report
Comments and Suggestions for Authors
This is a well-designed and well-written review of data obtained by database-search by search terms in Italian language. The findings and conclusions are promising as more and more plant species are now being studied for phyto-estrogenic effects and similar properties. I have no critical remarks. I can only encourage editors to proceed with this manuscript so that it can appear in this journal.
Author Response
authors thank reviewer for his positive comments